# Facilitatory rTMS over the Supplementary Motor Cortex Impedes Gait Performance in Parkinson Patients with Freezing of Gait

**DOI:** 10.3390/brainsci11030321

**Published:** 2021-03-03

**Authors:** Florian Brugger, Regina Wegener, Florent Baty, Julia Walch, Marie T. Krüger, Stefan Hägele-Link, Stephan Bohlhalter, Georg Kägi

**Affiliations:** 1Department of Neurology, Kantonsspital St. Gallen, CH-9007 St. Gallen, Switzerland; julia.walch@kssg.ch (J.W.); stefan.haegele-link@kssg.ch (S.H.-L.); georg.kaegi@kssg.ch (G.K.); 2Laboratory for Motion Analysis, Department of Paediatric Orthopaedics, Children’s Hospital of Eastern Switzerland, CH-9007 St. Gallen, Switzerland; reginawegener@gmx.de; 3Department of Orthopaedics and Traumatology, Kantonsspital St. Gallen, CH-9007 St. Gallen, Switzerland; 4Department of Pulmology, Kantonsspital St. Gallen, CH-9007 St. Gallen, Switzerland; florent.baty@kssg.ch; 5Department of Neurosurgery, Kantonsspital St. Gallen, CH-9007 St. Gallen, Switzerland; marie.krueger@kssg.ch; 6Department of Stereotactic and Functional Neurosurgery, University Medical Center Freiburg, D-79106 Freiburg, Germany; 7Neurocenter, Luzerner Kantonsspital, CH-6000 Luzern, Switzerland; stephan.bohlhalter@luks.ch; 8Department of Neurology, University Hospital Bern, Inselspital, University of Bern, CH-3000 Bern, Switzerland

**Keywords:** rTMS, freezing of gait, supplementary motor cortex, Parkinson’s disease

## Abstract

Freezing of gait (FOG) in Parkinson’s disease (PD) occurs frequently in situations with high environmental complexity. The supplementary motor cortex (SMC) is regarded as a major network node that exerts cortical input for motor control in these situations. We aimed at assessing the impact of single-session (excitatory) intermittent theta burst stimulation (iTBS) of the SMC on established walking during FOG provoking situations such as passing through narrow spaces and turning for directional changes. Twelve PD patients with FOG underwent two visits in the off-medication state with either iTBS or sham stimulation. At each visit, spatiotemporal gait parameters were measured during walking without obstacles and in FOG-provoking situations before and after stimulation. When patients passed through narrow spaces, decreased stride time along with increased stride length and walking speed (i.e., improved gait) was observed after both sham stimulation and iTBS. These effects, particularly on stride time, were attenuated by real iTBS. During turning, iTBS resulted in decreased stride time along with unchanged stride length, a constellation compatible with increased stepping frequency. The observed iTBS effects are regarded as relative gait deterioration. We conclude that iTBS over the SMC increases stepping frequency in PD patients with FOG, particularly in FOG provoking situations.

## 1. Introduction

Freezing of gait (FOG) in Parkinson’s disease (PD) is defined as an episodic inability of forward progression of the feet despite the intention to walk along with the subjective feeling of being glued to the floor [1]. It is often provoked when patients pass through narrow spaces or when they have to turn around their body axis to change walking direction [2]. FOG also features several interepisodic gait abnormalities such as disturbed bilateral leg coordination with higher gait asymmetry, increased stride time variability, or higher stepping frequency, particularly before the occurrence of FOG episodes [3,4,5].

FOG is the consequence of a transient breakdown of a large hierarchically organized cortical-subcortical network engaged in gait control [6,7]. The supplementary motor cortex (SMC) is presumably one of the key hubs within this network. It plays a central role in the planning and generation of self-paced movements, interlimb coordination, as well as voluntary suppression of unintended motor programs [8]. The SMC is assumed to keep the balance between motor initiation and motor suppression [9]. Its relevance for gait control is underpinned by the occurrence of FOG-like gait patterns in patients with SMC lesions [10,11]. It is not surprising that the SMC is considered to play a central role in PD and, in particular, in PD-related gait impairment.

Several methodological approaches were used in the past to assess SMC function in FOG: functional MRI studies revealed reduced SMC activation in PD patients with FOG during virtual reality gait tasks [12,13]. In another fMRI study, reduced SMC activation was demonstrated during episodic motor blocks in a stepping task, which was performed in the MR scanner. The motor blocks were considered as being similar to FOG episodes. [14]. We recently demonstrated that the so-called Bereitschaftspotential (BP) before self-initiated movements was smaller in PD patients with FOG compared to those without [15]. The BP presumably reflects preparatory cognitive processes in the SMC prior to voluntary movements [16]. Furthermore, we found stronger cortico-cortical coherence between the SMC in the beta frequency band of neuronal oscillation along with disturbed beta desynchronization prior to voluntary movements in patients with FOG compared to those without [15]. Beta oscillations arguably reflect synchronized neuronal activity at a frequency between 15 and 30 Hz and are modulated by voluntary movements. Cortical beta is desynchronized prior to the initiation of voluntary movements, whereas it increases during tonic position holding. Beta oscillations been proposed to preserve the current state of the motor system [17].

In several studies, repetitive transcranial magnetic stimulation (rTMS) over the premotor cortex including the SMC was evaluated with regard to its potential to improve gait and, more specifically, FOG in PD. rTMS is a non-invasive stimulation technique which allows modulating neuronal plasticity of cortical brain regions [18]. The results of these studies were inconsistent ranging from improvement to deterioration of gait performance. These studies did, however, differ in the rTMS paradigm applied, the methods used to assess gait performance, the measuring methods (in part only rough), and the medication state when rTMS was applied [19,20,21,22,23]. In our aforementioned study, we applied facilitatory rTMS over the SMC aiming at increasing neuronal plasticity, and investigated its impact on gait initiation by the virtue of a 3D gait analysis [15]. In contrast to our initial hypothesis, we then observed a deterioration of gait resulting in a relative reduction in stride time. This stimulation effect was widely lacking during straight walking without obstacles.

Gait initiation, however, differs from other FOG-provoking situations such as turning during walking or passing through narrow spaces. In the latter situations FOG occurs during an already established rhythmical, in part, automatized motor program, whereas the initiation of gait requires a more profound change of the motor state. Although FOG during passing through narrow spaces or turning occurs out of an automatized motor program, these situations have a high environmental complexity that presumably still require stronger cortical motor control.

In this study, we set out to investigate the impact of intermittent theta burst stimulation (iTBS), a facilitatory rTMS paradigm, over the SMC on automatized walking during FOG-provoking situations such as turning around or passing through narrow spaces. We hypothesized that facilitatory rTMS improved performance of established walking in FOG-provoking situations.

## 2. Methods

### 2.1. Participants

Twelve PD patients who suffered from predominant OFF-FOG (PD+FOG) were included in this study (Table 1). FOG was defined if patients had ≥1 point on item 3 of the Freezing of Gait Questionnaire (FOG-Q), a reliable and commonly used rating scale [24] and if a conclusive history in this regarded was provided according to a neurologist’s (F.B.) judgment. Participants had to have the capacity to walk along a trail of >10 m in length without assistance (i.e., Hoehn and Yahr stadium <4). Patients were excluded from the study if they had any contraindication for rTMS, if they had undergone deep brain stimulation in the past, or if they suffered from severe cognitive impairment (Mini Mental Status Examination <24, Frontal Assessment Battery <9). Furthermore, participants who suffered from other concomitant neurological conditions (e.g., stroke) or orthopedic conditions that could have interfered with their ability to walk (e.g., leg length discrepancy) were also not considered. All participants were naïve for rTMS. PD medication had to be unchanged for >4 weeks prior to inclusion and remain unchanged for the duration of the study. The study protocol was approved by the local ethics committee (EKSG-10/149).

### 2.2. Study Design

This study was designed as a proof-of-principle study with a single-blind, randomized crossover study design (1:1 randomization). Participants had to come in for two visits separated by 1–4 weeks. The visits were scheduled in the OFF phase (see Appendix A, for the definition of the OFF). Subjects were randomized as either receiving real iTBS at visit 1, followed by sham stimulation at visit 2, or receiving sham stimulation at visit 1 followed by real iTBS at visit 2. A three-dimensional gait analysis (3DGA) and the Unified Parkinson’s Disease Rating Scale (UPDRS) III were performed in all participants at each visit before and immediately after the stimulation (either iTBS or sham stimulation).

### 2.3. Clinical Assessment

The clinical assessment at screening included the UPDRS I-III, Hoehn and Yahr stadium, FOG-Q, Frontal Assessment Battery (FAB), and Mini Mental Status Examination (MMSE). Similar to the 3DGA, the UPDRS III assessments were done before and immediately after the stimulation part at each visit. The UPDRS examinations were recorded on video. Later, a blinded rater (J.W.) scored all UPDRS III items apart from rigidity by means of the video.

### 2.4. 3D-Gait Analysis

Self-reflecting markers (diameter: 14 mm) were fixed bilaterally according to the Plug-in Gait model over the second metatarsal bone, the heel and lateral malleolus. Participants walked barefoot at self-selected normal speed on a walkway (length: 10.5 m). The legs were stratified according to the body side that was more affected by PD symptoms. In the first parcour (P-I), patients had to walk straight ahead without any obstacles. In the second (P-II), patients had to pass through a narrow space built up by two obstacles. In the third (P-III), patients had to walk around two obstacles by turning towards the side more affected by PD symptoms, and in the fourth (P-IV) by turning towards the less affected body side. The parcours were repeated three times each before and after the stimulation session (iTBS, sham). In P-I, two stride cycles were analyzed after full walking speed had been reached, in P-II, two stride cycles while passing the obstacles, and in P-III and P-IV all stride cycles required a 360° turn around the obstacle. Marker trajectories were captured by eight infrared cameras (Vicon Oxford, Oxford Metrics Ltd., United Kingdom, 200 Hz) and modeled in Nexus 1.8.1. Offline data review and analysis were conducted in Polygon v3.5.1 (Oxford Metrics, United Kingdom). Since no clear FOG episodes were detected when gait was assessed by 3DGA, statistical analysis was focused on the interepisodical gait parameters. The following parameters were provided as an automatic output by Polygon: stride time, stride length, step time, step length, cadence, absolute and relative single and double leg support time (DLST), opposite foot off, opposite foot contact, and walking speed (the latter variable was calculated for each leg separately by dividing the distance from two subsequent heel strikes by the time required). Furthermore gait asymmetry was calculated as described previously [5].

### 2.5. Repetitive Transcranial Magnetic Stimulation

iTBS was applied over the left and right SMC by virtue of a figure-of-eight coil (MagstimPro, United Kingdom). The coil was placed tangentially to the skull with the handle orientated 45° in posterior and lateral direction. As a first step, the hot spot for the tibialis anterior muscle over the contralateral motor cortex was determined. The hot spot was defined as the location where the largest motor evoked potential (MEP) could be elicited. As a second step, the active and resting motor threshold (AMT, RMT) were determined bilaterally at the hot spot. The respective thresholds were defined as the stimulation intensity for eliciting a MEP of ≥0.1 mV in 5/10 trials. A coil orientation oblique to the midline was shown to be associated with the lowest motor thresholds for leg muscles [25]. As a third step, iTBS was applied. The stimulation site was located 3 cm rostral to the hot spot for the tibialis anterior in the same sagittal plane. Stimulation intensity was set at 100% AMT. This approach is a slight modification of the original protocol [26]. It was chosen to also sufficiently stimulate the deep medial aspect of the SMC. It is important to note that previous studies using iTBS protocols with higher pulse intensities have also proven efficacy [27]. AMT and RMT were comparable at both visits (Table 2). The hot spot as well as the respective motor thresholds were determined at each visit separately. Sham stimulation was applied through a specific sham coil that had the same appearance as the real TMS coil and mimicked magnetic stimulation (MagstimPro, United Kingdom). The 3DGA and UPDRS III were completed within 45 minutes after stimulation. This latency corresponds to the expected duration of the stimulation aftereffect [26].

### 2.6. Statistics

Statistical analysis was done in MATLAB (Natick, MA, USA) and IBM SPSS Statistics 20 (International Business Machines, Armonk, NY, US). In order to reduce the number of gait parameters to a few representatives, a principal component analysis was applied. A cut-off criterion of 80% of the cumulative explained variance was used to determine the number of retained principal components. Parameters of interest were selected according to their proximity to the eigenvectors. Differences of gait parameters between the different parcours were calculated by using a linear mixed model with parcour as fixed effect and subjects as random effect.

A linear mixed model was set up to investigate the iTBS effect on gait. The gait parameter of interest was defined as dependent, task-repetition (i.e., unspecific pre- vs. poststimulation effect), and stimulation (i.e., visits with iTBS vs. sham) as independent fixed effects. Furthermore, to account for the different arms in the crossover design, subjects and randomization were included as random effects. The model specification for the MATLAB protocol can be found in Appendix B. Task-repetition and task-repetition × stimulation interaction were reported (in the results mentioned as ‘interaction’). The significance level was set to *p* < 0.05.

## 3. Results

### 3.1. Selection of Gait Parameters

The principal component analysis revealed three factors explaining 94.1% of variance of gait parameters of the leg on the more affected side and 84.0% of the leg on the less affected side. The list of gait parameters was reduced to three representative gait parameters, namely stride length, stride time, and relative DLST. To increase the interpretability of the results, we decided post-hoc to include walking speed and gait asymmetry.

At baseline, i.e., before stimulation, stride time was longer when patients had to pass through narrow spaces compared to the other three parcours. The largest stride length values were observed during walking without obstacles compared to all other parcours. When patients passed through narrow spaces, stride length was longer than on any of the turning parcours. Relative DLST was longest on both turning parcours, compared to the other parcours. The lowest walking speed was measured on both turning parcours, whereas it was fastest during walking without obstacles and intermediate while passing through narrow spaces. Gait asymmetry was larger on the turning parcours than during walking straight ahead without any obstacle and while passing through narrow spaces. Details of the gait parameters are summarized in Table 3.

### 3.2. Impact of iTBS on Gait

#### 3.2.1. Walking without Obstacles (P-I)

A significant task-repetition effect led to an increase of stride length after stimulation on both legs, regardless of whether iTBS or sham stimulation was applied (more affected leg: 0.05 m; (95% confidence interval: 0.02–0.12); *p* = 0.010; less affected leg: 0.07 m (0.02–0.12); *p* = 0.009). This task-repetition effect was attenuated by trend by iTBS (interaction: more affected leg: −0.03 m (−0.06–0.00); *p* = 0.068; less affected leg: −0.03 m (−0.06–0.00); *p* = 0.057). Walking speed increased on the less affected side after the stimulation block (task-repetition: 0.09 m/s (0.02–0.16); *p* = 0.011) and by trend on the more affected side (task-repetition: 0.07 m/s (0.0–0.14); *p* = 0.054). There was a tendency towards an attenuation of this increase after iTBS on the less affected leg (interaction: −0.04 (−0.08–0.00); *p* = 0.051).

#### 3.2.2. Passing through Narrow Space (P-II)

Stride time decreased after the stimulation block on both legs (task-repetition: more affected side: −0.09 s (−0.14–(−0.05)); *p* < 0.001; less affected side: −0.08 s (−0.13–(−0.04)); *p* < 0.001), but this decrease was less prominent after iTBS on the more affected side (interaction: 0.03 s (0.01–0.06); *p* = 0.020) and borderline on the less affected side (interaction: 0.03 s (−0.01–0.06); *p* = 0.063). Walking speed also increased on both legs after stimulation (task-repetition: more affected side: 0.13 m/s (0.05–0.21); *p* = 0.002; less affected side: 0.15 m/s (0.07–0.23); *p* < 0.001), but again this effect was attenuated after iTBS on the less affected side (interaction: −0.06 m/s (−0.11–(−0.01)); *p* = 0.020) and borderline on the more affected side (interaction: −0.05 m/s (−0.10–0.00), *p* = 0.059). Stride length became longer on the less affected side after stimulation (task-repetition: 0.08 m (0.02–0.14); *p* = 0.012) with a trend towards a less prominent increase after iTBS (interaction: −0.03 m (−0.07–0.01); *p* = 0.088). Furthermore, there was a significant decrease of relative DLST after stimulation on the more affected leg (task-repetition: −3.72 (−6.59−(−0.85)); *p* = 0.011); again with a tendency towards a less prominent decrease after iTBS (interaction: 1.56 (−0.25–3.38); *p* = 0.091).

#### 3.2.3. Turning towards the Side More Affected by PD (P-III)

There was a slight trend towards increased stride time after the stimulation block on the less affected side (task-repetition: 0.09 s (−0.01–0.19); *p* = 0.089). After iTBS, a significant decrease of stride time could be observed on both sides (interaction: more affected side: −0.07 s (−0.13–(−0.01)); *p* = 0.033; less affected side: −0.08 s (−0.14–(−0.02)); *p* = 0.007). Furthermore, there was an overall trend towards larger gait asymmetry after the stimulation block (14.92 (−0.34–30.19); *p* = 0.055), but gait asymmetry tended to decrease after real iTBS (−8.54 (−17.50–0.43); *p* = 0.062).

#### 3.2.4. Turning Away from the Side More Affected by PD (P-IV)

There was a trend towards decreased stride time on the more affected leg after iTBS (interaction: −0.05 s (−0.11–0.01); *p* = 0.099).

The results for stride time for all four parcours are shown in Figure 1.

#### 3.2.5. Overall Motor Performance

There was neither a task-repetition nor task-repetition × stimulation effect on the UPDRS III scores, regardless of the blinding of the raters (*p* > 0.2; details see Appendix A).

## 4. Discussion

In this proof-of-principle study, we observed iTBS specific effects on spatiotemporal gait parameters. These effects were more prominent in FOG-provoking situations than during walking without obstacles. When patients passed through narrow spaces, a decrease of stride time paralleled by an increase of stride length and walking speed was observed after the stimulation. These changes were observed after both real iTBS and sham stimulation, and were interpreted as a relative improvement of gait. This effect may owe to the fact that the task was repeated within a short time period and thus points towards a learning effect. However, this task repetition effect on stride time and walking speed while passing through narrow spaces was attenuated by real iTBS, indicating relative gait deterioration.

On the turning tasks, such a learning effect was absent. Instead, iTBS resulted in a decrease of stride time, while walking speed and stride length remained unchanged. The latter combination of gait parameters is compatible with an increase of stepping frequency. Increased stepping frequency along with constant or decreased stride length is a gait pattern, which is observed immediately before actual FOG episodes [3]. Beyond this, there was a trend towards a decrease of gait asymmetry after iTBS in the parcour, in which patients turned towards the more affected side. An increase of gait asymmetry, however, is required to effectively accomplish a turning task [28]. The finding of a relative decrease of gait asymmetry in this situation can thus be discussed in the context of disturbed bilateral leg coordination, which has been described in PD patients with FOG [5,29,30]. 

All together, the observed alterations of the gait pattern may hence also be regarded as relative gait deterioration. It should be noted that, although in all parcours a relative deterioration of gait was encountered after iTBS, stride time changed, in relation to sham stimulation, in opposite directions while passing through narrow spaces and while turning, respectively. This observation may be explained by a non-linear relationship of the different spatiotemporal gait parameters. A further explanation for the different direction of these changes can be diverging control of gait parameters in the different FOG-provoking situations. The latter explanation is also supported by the observation that at baseline, stride time was the longest while passing through narrow spaces and that the strongest learning effect was seen in this parcour, whereas the learning effect was not observed in the turning parcours. During walking without FOG-provoking situations, mainly stride length and walking speed increased as the consequence of task repetition, but there was only a weak iTBS effect on gait, which did not reach statistical significance. We postulate that the iTBS effects are specific for the gait domain, since overall motor performance did not change after stimulation. Previous studies that assessed the impact of rTMS of the SMC on gait in PD yielded inconsistent results [19,20,21,23]. In two recent studies, it was shown that the application of facilitatory rTMS of the SMC in the ON led to improved gait on standardized walking tasks and reduced FOG severity [21,23]. These results are in contrast to our findings. One explanation could be the different medication states when rTMS was applied. Furthermore, the rTMS protocols applied were different to ours. Our results are, however, congruent with our previous study in which we demonstrated a detrimental effect of single-session iTBS of the SMC on gait initiation [15].

The neurophysiological basis of the stimulation effects observed in this study remains speculative. We propose that the SMC exerts, in part, its function by the modulation of inhibitory efferents, e.g., through the hyperdirect pathway which projects on the subthalamic nucleus [9]. In the context of inhibitory circuitries, beta oscillations are of great interest because they are considered to preserve the current motor state [17]. On the other hand, they need to be desynchronized prior to the voluntary initiation of a new motor program [31,32]. There are first indications that beta also plays a central role in FOG [15,33,34]. We speculate that the facilitatory rTMS paradigm applied during rest and in the state off dopaminergic medication in our study could have reinforced these beta oscillating inhibitory circuitries. In contrast to automatized gait on a plain surface without any obstacles, FOG-provoking situations often require a flexible adaptation of gait metrics. Reinforcement of inhibitory circuitries, however, may hinder this flexible adaptation. Interestingly, the strongest modulatory effect of iTBS was seen on stride time. This observation may be in line with the view that the SMC is strongly involved in internal time keeping as well as encoding and maintaining rhythms [35,36]. In this regard, it is also worth noting that faulty time scaling is a core feature of FOG reflected by impaired regulation of stride variability and increased stepping frequency preceding FOG episodes [3,4].

Our study has some limitations. First, we did not gather information on the patients’ impression of whether gait has changed after stimulation. This information would have allowed better discussion regarding the clinical relevance of the iTBS effects. Second, several iTBS-related effects were only borderline statistically significant. One explanation for this is the low number of patients included in this study. Furthermore, we did not apply corrections for multiple comparisons. It should be noted that this study was designed as a proof-of-principle study, which justified a smaller number of participants and abstaining from correcting for multiple comparisons in the first instance. However, the data need to be confirmed in future studies. Third, we did not observe a clear FOG episode in any of our patients, although all participants suffered from FOG. It is well known that FOG occurs much less frequently in an experimental than in a domestic environment [37], rendering measurement of FOG in a gait lab tricky. Instead of actual FOG episodes, we focused on the more robust interepisodical gait parameters. It may, however, be speculated that providing additional cognitive overload may have put patients at higher risk to freeze during the experiments [38]. Fourth, by using higher AMT in our iTBS protocol, we expected to stimulate also the parts of the SMC located deep in the interhemispheric fissure. Although stimulation in the depth can be achieved by a figure-of-eight coil, a double-cone coil would have been more suitable for this purpose due to its stronger magnetic field with higher penetration depth [39].

## 5. Conclusions

Facilitatory iTBS over the SMC modulated spatiotemporal gait parameters, particularly stride time, in PD patients with FOG in FOG-provoking situations. The overall effects can be regarded as a relative deterioration of gait, mainly in the time domain. Further studies are needed to investigate the impact of the different rTMS protocols on neuronal oscillations in the SMC in PD.

## Figures and Tables

**Figure 1 brainsci-11-00321-f001:**
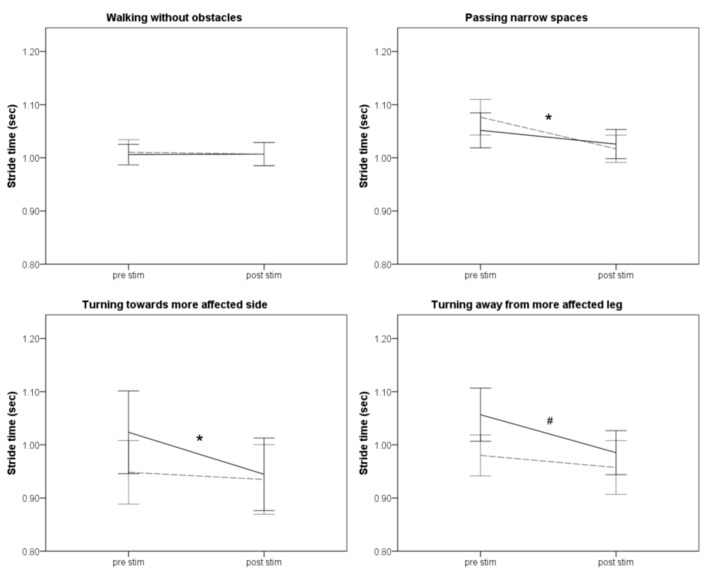
The results for stride time on the leg on the body side more affected by Parkinson’s disease (PD) symptoms are shown (mean ± 95% confidence interval). The results from the visit when intermittent theta burst stimulation (iTBS) was applied are shown as solid black line, those from the visit when sham was applied as dashed gray line. The asterisk (*) indicates statistically significant task-repetition × stimulation interactions (i.e., iTBS specific effects; *p* < 0.05), the hash (#) borderline significant interactions (*p* < 0.1).

**Table 1 brainsci-11-00321-t001:** Demographics and disease characteristics.

Number of Participants	12
Age (years)	64.30 (52.8–68.3)
Sex (male / female)	10/2
Disease duration (years)	12.5 (10.5–15.0)
Hoehn and Yahr	2.0 (2.0–2.8)
UPDRS I	3.0 (0.5–4.0)
UPDRS II	16.5 (10.3–20.8)
UPDRS III (OFF) (blinded)	30.0 (25.3–33.8)
UPDRS III (OFF) (unblinded, incl. rigidity item)	37.0 (31.0–43.8)
FOG-Q	10.5 (10.0–11.8)
FOG duration (years since symptom onset)	3.0 (1.0–8.0)

Note: Data are reported as medians (interquartile range). Abbreviations: FOG—freezing of gait, FOG-Q—Freezing of Gait Questionnaire, UPDRS—Unified Parkinson’s Disease Rating Scale.

**Table 2 brainsci-11-00321-t002:** Minimal stimulation intensities to elicit motor evoked potentials (MEP).

	Visit with Sham Stimulation	Visit with Real iTBS	*p*-Value
Hemisphere contralateral to the clinically more affected side			
RMT	86.25 ± 9.78	87.50 ± 11.51	0.78
AMT	73.67 ± 6.17	73.00 ± 8.56	0.83
Hemisphere contralateral to the clinically less affected side			
RMT	83.92 ± 15.24	82.67 ± 14.46	0.84
AMT	70.33 ± 10.45	70.25 ± 10.37	0.99

Note: The table shows the average stimulation intensity to elicit a MEP of >0.1 mA in five out of ten trials at the anterior tibialis muscle during rest (= resting motor threshold, RMT) and during slight activation of the muscle (= active motor threshold, AMT) at each visit. The data (mean ± standard deviation) are provided as percentage of the maximum output of the stimulator. Abbreviations: AMT—active motor threshold; iTBS—intermittent theta-burst stimulation; MEP—motor evoked potentials; RMT—resting motor threshold.

**Table 3 brainsci-11-00321-t003:** Gait parameters at baseline.

Gait Parameters	Walking without Obstacles	Passing through Narrow Spaces	Turning towards More Affected Side	Turning towards Less Affected Side	*p*-Value
**more affected side**					
stride time (s)	1.01 ± 0.06	1.06 ± 0.10	0.99 ± 0.21	1.02 ± 0.14	0.007
stride length (m)	1.13 ± 0.18	1.07 ± 0.18	0.38 ± 0.14	0.40 ± 0.15	<0.001
DLST (%)	30.04 ± 4.48	31.44 ± 5.24	41.16 ± 7.22	38.97 ± 6.47	<0.001
walking speed (m/s)	1.13 ± 0.20	1.02 ± 0.21	0.38 ± 0.11	0.39 ± 0.14	<0.001
**less affected side**					
stride time (s)	1.01 ± 0.06	1.06 ± 0.10	0.99 ± 0.22	1.01 ± 0.12	0.017
stride length (m)	1.13 ± 0.18	1.07 ± 0.18	0.38 ± 0.16	0.39 ± 0.10	<0.001
DLST (%)	30.17 ± 4.39	31.49 ± 4.96	41.98 ± 7.56	39.26 ± 7.06	<0.001
walking speed (m/s)	1.12 ± 0.20	1.01 ± 0.21	0.38 ± 0.15	0.39 ± 0.10	<0.001
**both legs**					
gait asymmetry (%)	4.61 ± 5.20	5.85 ± 6.46	15.93 ± 11.40	15.53 ± 13.96	<0.001

Note: The table shows the gait parameters at baseline (i.e., prior to the stimulation block) for each leg. Six values were available for each subject i.e., three measurements at each visit. Values are provided as mean ± standard deviation of these six measurements. *P*-values derive from the linear mixed model used to compare gait performance across the four parcours.

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
