# Peer review of "Facilitatory rTMS over the Supplementary Motor Cortex Impedes Gait Performance in Parkinson Patients with Freezing of Gait"

_brainsci, 2021, doi:10.3390/brainsci11030321_

Round 1
Reviewer 1 Report
Thank you for inviting me to review this manuscript by Brugger and colleagues, in which the authors used intermittent Theta Burst Stimulation (iTBS) over the bilateral supplementary motor area (SMA) of the cerebral cortex in 12 individuals with Parkinson’s disease and self-reported freezing of gait in the dopaminergic OFF state. The authors observed weak evidence to suggest that iTBS of the SMA decreased freezing-like behaviour in some instances (e.g., lessened impairments association with walking through narrow spaces), however the effects were also difficult to parse from sham on a number of trials.
My primary concern with this manuscript is that these issues were all compounded by a relatively small sample-size, which I believe impeded the authors ability to conduct well-powered statistical analyses of their data. Having said that, I can appreciate the challenges associated with recruiting and testing individuals with PD and FOG in the OFF state. At the least, I caution the authors to carefully assess their data for features of non-Gaussian distributions, as the paroxysmal nature of freezing can often (at least in our hands) lead to the presence of rare-events (hence, stretched tails of statistical distributions) that also happen to characterise the very phenomenon of interest.
I have added some notes below that I hope will help the manuscript.
Major
- A sample size of 12 is insufficient for the statistics required to test the authors hypothesis. I recommend that the authors conduct a power analysis and then re-assess their ability to test their hypotheses in these data. At the very least, this feature of the current study should be added to the Limitations section of the Discussion.
- Given the paroxysmal nature of freezing, the authors may wish to consider non-parametric statistics.
- P2, line 59: The comment: “Beta oscillations are regarded as a local idling rhythm, which preserves the current motor state.” should be defended with a reference, particularly as it’s not the typical definition ascribed to beta oscillations.
Minor
- There are numerous instances in which a hyphen (“-“) is entered into the middle of words, perhaps from a previous format?
- P2, line 53 – 2x typo: “step-ping task akin freezing m.”
- Sections of the Introduction and Discussion are not paragraphed effectively, leaving the reader with a block of text that makes interpretation challenging.
Reviewer 2 Report
In this proof-of-principle study, Brugger and colleagues analyse the effects of intermittent theta burst rTMS over the supplementary motor cortex on freezing of gait in patients with Parkinson's disease. The authors observe that facilitatory rTMS seem to deteriorate gait parameters compared to sham stimulation. This study has several limits, particularly sample size and study design (single-blinded study), that the authors acknowledge in the discussion section of the manuscript. I have the following comments for the authors to take into account:
- Title: add "of gait" after "freezing".
- Line 53: revise the content because the phrase is difficult to understand.
- Line 95: patients with dementia are not classified by FAB or MMSE scores but by loss of autonomy in activities of daily living. Cognitive decline may be classified by MMSE. Please revise.
- Table 1: please report only median and interquartile range (lower quartile-upper quartile). Including also median and sd is confusing.
- Study design: did the authors correct for the order in which stimulation was performed?
- Line 121: what does the last phrase mean? Please clarify.
- Did the authors perform correction for multiple comparisons? Considering the series of tests performed they should state so and if not performed, why.
- Figure 1. Perhaps adding * to significant differences would help the reader.
Round 2
Reviewer 1 Report
The authors have adequately addressed my concerns.
Author Response
As far I can see here there are no issues left which need to be addressed. Am I right?